# Holocene evolution of the North Atlantic subsurface transport

Janne Repschläger [1, a], Dieter Garbe-Schönberg [1], Mara Weinelt [2], and Ralph Schneider [1]

[1]Institute of Geosciences, Kiel University, 24118 Kiel, Germany
[2]Institute of Prehistoric and Protohistoric Archaeology, Kiel University, 24118 Kiel, Germany
[a]*now at* Climate Geochemistry Department, Max-Planck-Institute for Chemistry, 55128 Mainz, Germany

*Correspondence to*: Janne Repschläger (J.Repschlaeger@mpic.de)

**Abstract.** Previous studies suggested that short term freshening events in the subpolar gyre can be counterbalanced by advection of saline waters from the subtropical gyre and thus stabilize the Atlantic Meridional Overturning Circulation (AMOC). However, little is known about the inter-gyre transport pathways. Here, we infer changes in surface and subsurface
transport between the subtropical and polar North Atlantic during the last 11000 years, by combining new temperature and salinity reconstructions obtained from combined $\delta^{18}0$ and Mg/Ca measurements on surface and subsurface dwelling foraminifera with published foraminiferal abundance data from the subtropical North Atlantic, and with salinity and temperature data from the tropical and subpolar North Atlantic. This compilation implies an overall stable subtropical warm surface water transport since 10 ka BP. In contrast, subsurface warm water transport started at about 8 ka but still with
subsurface heat storage in the subtropical gyre. Full strength of intergyre exchange was probably reached only after the onset of northward transport of warm saline subsurface waters at about 7 ka associated with the onset of the modern AMOC mode. A critical evaluation of different potential forcing mechanisms leads to the assumption that freshwater supply from Laurentide ice sheets was the main control on subtropical to subpolar ocean transport at surface and subsurface levels.

## 1 Introduction

The Holocene, though often considered a generally stable warm climate mode, is characterized by distinct long-term climate trends, superimposed by strong oscillations on millennial to decadal time scales (e.g. Mayewski et al. 2004). The long-term evolution is formally divided into the Early Holocene (11.6 to 8.2 ka BP), the Mid Holocene (8.2 to 4.3 ka BP) and the Late Holocene (4.3-0 ka BP) (Walker et al., 2012). This gradual change is caused by variations in the incoming solar radiation
due to changes in Earth's orbit, expressed as insolation at 60°N in June. Northern hemisphere insolation changes are inferred to control strength and position of the global wind systems, and to have caused the early Holocene thermal maximum observed in the high northern latitudes (e.g. Leduc et al., 2010; Moros et al., 2006). This warming induced the final melting of glacially extended ice sheets and released a significant amount of meltwater from the Laurentian ice sheets into the North Atlantic (e.g. Jennings et al., 2015).
Shorter periodic climatic cycles, with average duration of about 1500 years, are observed in many Holocene climate records, and are superimposed on the long-term trend. The causes of these cycles are controversially discussed, with hypotheses

ranging from meltwater pulses induced by variations in solar irradiance, to internal ocean-ice-atmosphere feedback mechanisms, or volcanism (e.g. Andrews and Giraudeau, 2003; Bond et al., 2001; Campbell et al., 1998; Schulz et al., 2004; Viau et al., 2006; Wanner, 2008). Due to the wide range of feedback mechanisms, involving northern hemisphere oceanic and atmospheric circulation, climate patterns with strong local differences are observed for the Holocene. Therefore,

identification and understanding of the importance of the different driving forces and stabilizing mechanisms for the Holocene climate remains challenging. It is assumed that the stabilization is related to a strong Atlantic Meridional Overturning Circulation (AMOC), driven by deep-water formation in the Nordic Seas and the Labrador Sea, and fuelled by the northward transport of warm saline waters. In this scenario, the main stabilization occurred via the formation of Labrador Sea Water (LSW), which was probably well established at about 7 ka BP (Hoogakker et al., 2011; Kissel et al., 2013;

Solignac et al., 2004; Thornalley et al., 2013). A stable LSW formation relies on the interaction between warm water transport in the clockwise circulating subtropical gyre (STG) and the subpolar gyre (SPG) circulating counter-clockwise in the Irminger Basin and the Labrador Sea (Thornalley et al., 2009). Anti-phase heat and salt storage within the STG during times when SPG freshens (Cléroux et al., 2012; Thornalley et al., 2009) and the later advection of the warm saline water into the SPG leads to a strengthening of the deepwater formation by convection within the SPG (e.g. Born and Stocker, 2014).

The warm water transport into the subpolar North Atlantic and the Nordic Seas via surface flow within the NAC is relatively well known from oceanographic studies (Sarafanov et al., 2012) and paleoceanographic reconstructions (e.g. Andersen et al., 2004; Came et al., 2007; Farmer et al., 2011; Giraudeau et al., 2010; Moros et al., 2004; Staines-Uréas et al., 2013). However, investigation of different pathways, surface and subsurface, of the warm water masses from the tropical into the subpolar North Atlantic is limited to a few studies thus far, as is true also for the different branches of the warm water

transport either into the Azores or the Irminger Current, the latter feeding into the Labrador Sea.(Figure 1).

Drifter and modelling studies of the modern ocean indicate a lack of warm surface water transport *vis-a-vis* a major warm water transport into the SPG via a subsurface (>500m) pathway into the Irminger Sea (Brambilla and Talley, 2006; Foukal and Lozier, 2016; Lozier, 2012). In contrast, hydrographic data from ocean cross sections indicate a transport of warm, saline water between 10 and 1000 m water depth into the Irminger Basin and Labrador Sea (Sarafanov et al., 2012; Våge et

al., 2011) with a recirculation and mixing zone of NAC waters into the SPG in the North west corner of the SPG (Mertens et al., 2014).

Most paleo reconstructions focus on the surface water structure in the subpolar North Atlantic (Came et al., 2007; Farmer et al., 2011; Staines-Uréas et al., 2013) and neglect the probably significant subsurface component of the transport. The two studies that investigated the interaction between the STG and SPG (Solignac et al., 2004; Thornalley et al., 2009) address

surface water pathways mainly, while changes in the tropical origin of the water masses are not considered. Thus a systematic investigation of the subsurface transport from the tropical into the subpolar North Atlantic and its potential impact on the stabilization of the Holocene AMOC mode is apparently missing. For that reason, the role of surface and subsurface warm water transport from the tropics into the subpolar North Atlantic on the evolution and stabilization of the Holocene climate mode is investigated and different forcing mechanisms are discussed here. This knowledge is pivotal to establish

more robust scenarios for future AMOC changes under ongoing anthropogenic climate change leading to warming and potential freshening of the subpolar NA.

## 2 Regional Setting

The warm water circulation in the North Atlantic is governed by the North Atlantic Current (NAC) that transports warm saline water (e.g. Roessler et al., 2015) into the high northern latitudes. On its northward path, several warm water currents branch off the main pathway (Figure 1). The most distinct ones are the Azores Current (AC) and the Irminger Current (IC). The eastward flowing AC leaves the NAC between 30 and 40°N and forms the northern boundary of the STG. The latter rotates clockwise and recirculates warm water between 10 and 40°N. The AC is associated with the Azores Front (AF) that is characterized by strong eddy activity and forms the boundary between cool transitional and warm oligotrophic subtropical waters. The formation and about 75% of volume transport with the AC is driven by the strength of Mediterranean outflow (Volkov and Fu, 2010), whereas 25% of the AC strength and the latitudinal position of the AF and AC west of 20°W are probably controlled by the position of the westerly wind belt  (Volkov and Fu, 2011) with increasing/decreasing wind strength leading to a southward/northward displacement of the AF, respectively. South of Iceland at about 50°N, the IC branches off the main northward path of the NAC. It flows in north-westward direction following the bathymetry of the Irminger Basin. South of the Denmark Strait, The IC returns southward following the coast of Greenland into the Labrador Sea (Spall and Pickart, 2003). It thereby feeds warm saline waters into the counter-clockwise rotating SPG, and fuels deepwater convection in the Labrador Sea (Born and Stocker, 2014; Sarafanov et al., 2012; Våge et al., 2011).

## 3 Methods

For our study, we used core GEOFAR KF16 taken at 38°N south of the Azores Islands from 3060 m water depth at the Mid Atlantic Ridge. The age model for this core was published in (Schwab et al., 2012). The position of core GEOFAR KF16 (38°N, 31.13°W) in the vicinity of the AF is ideal to trace changes in STG position and associated varying influence of subtropical and temperate waters (Repschläger et al., 2015). Results are compared to published data from the western subtropical (Bahr et al., 2013; Cléroux et al., 2012) and subpolar North Atlantic (Thornalley et al., 2009) along the warm water route.

We combine published $\delta^{18}O$ data of planktonic surface and subsurface dwelling foraminifera *G. ruber* w. (Schwab et al., 2012) and *G. truncatulinoides* (Repschläger et al., 2015) with new Mg/Ca records from the same species. All measurements were conducted on samples of the size fraction 315-400 µm on mono specific samples of 10-25 specimens of *G. ruber* w. and 15 specimens of mixed left and right coiling *G. truncatulinoides*. Stable isotope analyses were carried out at Leibniz Laboratory for Radiometric Dating and Stable Isotope Research at Kiel University. For analyses a Finnigan MAT 523 mass spectrometer coupled with a Kiel I carbon preparation device was used and results were calibrated to the Vienna Vienna Pee

Dee Belemnite (V-PDB) scale. The 2 σ standard deviation (std) obtained from 10 replicates of downcore samples was +/-0.11 ‰ for *G. ruber* w. 0.12 ‰ for *G. truncatulinoides*. Mg/Ca analyses followed the cleaning procedure of Martin and Lea (2002), using reductive and oxidative cleaning. Measurements were carried out with a simultaneous inductively coupled plasma-optical emission spectrometry (ICP-OES) instrument with radial plasma observation. Potential shell contamination or coatings by authigenic minerals were monitored using additional Fe, Al and Mn measurements. Analytical 2 σ std was +/- 0.20 mmol/mol Mg/Ca for *G. ruber* w. and +/-0.14 mmol/mol Mg/Ca for *G. truncatulinoides*. To convert Mg/Ca values into water temperature estimates, we used the principal equation format Mg/Ca = b exp (a*T) with species-specific variables a=0.76 and b=0.07 for *G. ruber* w. and a=0.78 and b=0.04 for *G. truncatulinoides* proposed for the equation for mixed subsurface dwellers (Cléroux et al., 2008). The summed analytical and calibration 2 σ std is +/- 1°C for *G. ruber* w. and +/- 2°C for *G. truncatulinoides*. The calibrated Mg/Ca temperatures estimates for both species in core top samples match well with modern surface and subsurface (200 m depth) temperatures at the Azores coring site. Seasonal temperature effects on the foraminiferal Mg/Ca signal are assumed to play a subordinate role at our coring site (Repschläger et al., 2015). In addition, we assume that the subsurface temperature signal of *G. truncatulinoides* is predominantly determined by the conditions at the AF and not by the migration of *G. truncatulinoides* to shallower, warmer water depths or by thermocline shoaling (see supplementary information and (Repschläger et al., 2015)).

Changes in salinity are reconstructed following the procedure described in Repschläger et al. (2015). The temperature effect, estimated from the Mg/Ca was removed from the foraminiferal $\delta^{18}O_{carbonate}$ using the general equation of *Shackleton (1974)* for both, *G. ruber* w. and *G. truncatulinoides* in order to be consistent with the datasets used for comparison. A correction for VPDB to SMOW was included. Resulting seawater $\delta^{18}O_w$ values are discussed in the following. For estimation of $\delta^{18}O_w$ uncertainties we followed the approach of Cleroux et al., (2011) and used their equation S8

$$\sigma_{\delta^{18}O_{sw}} = \sqrt{(\sigma_{\delta^{18}O_{foram}})^2 + \sigma_{Temp}^2(0.23)^2}$$

leading to a calculated 2 σ std of +/- 0.35 ‰ for *G. ruber* w. and +/- 0.68 ‰ for *G. truncatulinoides,* respectively. For interpretation of the temperature and $\delta^{18}O_w$ values, we only used 3-point average time series to investigate longer-term trends in the datasets. The $\delta^{18}O_w$ records were corrected for ice volume, using the relative sea level composite curve of (Austermann et al., 2013) assuming a 1.2 ‰ glacial-interglacial change in marine $\delta^{18}O$. The corrected data are expressed as $\delta^{18}O_{w-ivc}$ values throughout the text.

Changes in the AF front were reconstructed using the relative abundance of *G. ruber* w. published by (Weinelt et al., 2015) These abundance counts have an 2 σ std of +/- 2.5 %. As *G. ruber* w. is most abundant within the STG (Ottens, 1991; Schiebel et al., 2002; Storz et al., 2009), low/high abundances therefore indicate a southward/northward movement of the AF relative to the coring site. Because the position of the AF is related to changes in the westerly wind belt, the abundances of *G. ruber* w. indicate the relative contribution of subtropical water and can be used as tracer for the position of the northern STG rim and thus of the position of westerlies (see also argumentation in Repschläger et al., 2015).

## 4 Results

Surface and subsurface $\delta^{18}O$ records (Figure 2a) show a parallel trend over the Holocene, with a 0.5 ‰ decrease from 0.2 ‰ and 1.5 ‰ at 11 ka BP to -0.3 ‰ and 0.9 ‰ at 8 ka BP, respectively. After 6 ka BP both records stabilize with minor fluctuations of 0.2 ‰ around an average of 0.3 ‰ and 1 ‰, respectively. Additionally, both records show a major positive $\delta^{18}O$ excursion of 0.7 ‰ between 7 and 6 ka BP that is probably related to a discontinuity within the core.

The SST record of *G. ruber* w. (Figure 2b) is relatively stable over the entire Holocene, fluctuating between 17.5 and 20°C. In contrast, subsurface temperatures ($T_{sub}$) show an increase from low temperatures around 12.5°C at 11 ka BP to about 17°C between 11 and 8 ka BP. Over the last 7 ka BP, $T_{sub}$ fluctuate by 2.5°C around an average value of 16°C and parallel the surface water record with an average offset of ~ 4°C.

The $\delta^{18}O_{w-ivc}$ records of the surface and subsurface water (Figure 2c) both show strong short-term variability of 0.5 ‰. The 5-point average values of both records, however, reveal an analogous evolution (Figure 3). Both records are similar between 11 and 10.5 ka with average values between 0.3 ‰ to 0.5 ‰. Between 10.5 and 10 ka BP the records start to increase to values between 1.5 ‰ and 1.7 ‰ at 6 ka BP. After 6 ka BP surface and subsurface $d^{18}O_{w-ivc}$ records separate, but only slightly exceeding the error of calculation.

Low abundances of *G. ruber* w. <15% are found in sediments older than 9 ka BP. High abundances of *G. ruber* w. (~20 %) are observed between 8 and 4 ka BP. Between 6 and 7 ka BP a short excursion with a decrease in *G. ruber* w. abundances to as low as 10% coincide with an increase in the $\delta^{18}O$ records and a decrease in $T_{sub}$. This event is associated with a discontinuity in the sediment core. After 4 ka BP the abundance of *G. ruber* w. decreases to 15%.

## 5 Discussion

The AF is characterized by the depth of the 15°C isotherm (Gould, 1985) and is therefore not traceable in SST (Alves et al., 2002) but by distinct shifts in foraminiferal assemblages from subtropical to transitional/subpolar species (Ottens and Nederbragt, 1992; Schiebel et al., 2002) and thus also by abundances of *G. ruber* w. (Repschläger et al., 2015 and citations therein). Low abundances of *G. ruber* w. (7- 15 %) (Figure 2d) between 11 and 9.5 ka BP indicate a southward displacement of the AF and correlate with low $T_{sub}$ and low subsurface $\delta^{18}O_{w-ivc}$. High abundances of *G. ruber* w. (~20 %) between 8 and 4 ka BP indicate a northward movement of the AF, coeval to a warming of the subsurface waters and high $\delta^{18}O_{w-ivc}$ in *G. truncatulinoides*. After 4 ka BP, the abundances of *G. ruber* w. decrease to about 15%, indicating a southern position of the AF. This decrease initially coincides with a slight decrease in surface and subsurface $\delta^{18}O_{w-ivc}$ between 4 and 3.5 ka BP but no temperature change is apparent for this interval.

Based on the records presented here, the Holocene can be divided into four sections: The Early Holocene (11-8.2 ka BP) is characterized by a thermal difference between surface and subsurface waters. Together with the quite similar values in the $\delta^{18}O_{w-ivc}$ records, this points toward a temperature-driven stratification during a period when the AF was south of its modern

position. In the early Mid Holocene (8-6 ka BP) the thermal difference between surface and subsurface waters decreased while the difference in $\delta^{18}O_{w-ivc}$ remained small but both records shift to slightly higher values. This phase can be interpreted as an intermediate phase with a weak thermal stratification but increasing salinities that is accompanied by a northward movement of the AF, reaching its northernmost position. In the late Mid Holocene (6-4 ka BP) the AF remained at the northernmost position while evolving differences between surface and subsurface T and S indicate a stabilization of the thermohaline stratification. The Late Holocene (4-0 ka BP) differs from the previous phase mainly in the position of the STG. The latter is replaced to a more southern position that is also observed under modern conditions.

**5.1 Transport between tropical and subpolar North Atlantic**

In order to investigate how changes in the mixed layer at our subtropical coring site can be related to changes in surface and subsurface warm water transport between the subtropical and subpolar North Atlantic, we compare our records with ODP 1058 and MD99-2203 records from the western subtropical (Bahr et al., 2013; Cléroux et al., 2012) and RAPID 12-1 from the subpolar NA (Thornalley et al., 2009). To infer to any connection to north- western SPG recirculation our new datasets are also compared to temperature reconstructions from HU-90-13-13 (P-013) and HU-91-045-094 (P094) from the Greenland Rise and Orphan Knoll (Solignac et al., 2004).

In general, SST estimates from the western subtropical, subtropical Azores and subpolar sites (Figure 3) fluctuate by 2°C around rather stable mean values of 27°C, 18°C and 10°C, respectively. The $\delta^{18}O_{w-ivc}$ records are fluctuating by +/- 0.4 ‰ around values of 1.2 ‰, 0.9‰, and 0 ‰. These fluctuations are within the errors of Mg/Ca temperature estimation and seawater $\delta^{18}O$ determinations. Assuming that the surface water properties at the three coring sites are mainly driven by changes in the surface water transport, we conclude that the mean northward surface water transport remained stable over the Holocene, associated with shifts in the AF latitudinal position.

Thus any variability in North Atlantic inter-gyre heat and salt exchange must originate from changes in the subsurface transport. The subsurface signal from the tropical site is rather constant during the entire Holocene, with average tropical $T_{sub}$ fluctuating by +/- 3°C around a mean of 17.5°C and $\delta^{18}O_{w-ivc}$ varying by 0.5 ‰ around a mean of 1.2 ‰, and with no apparent long-term trends. In contrast, distinct temporal differences are observed in the subsurface records at the subtropical and subpolar site (Figure 3b).

From 11 to 8 ka BP, coincident with a southern position of the AF, subtropical subsurface records are similar to subpolar values with low $T_{sub}$ of 12 and 10°C and low subsurface $\delta^{18}O_{w-ivc}$ of about 0.5 ‰. The surface water records suggest active northward water transport, similar to the state described at the transition between the Allerød and the Younger Dryas (Repschläger et al., 2015). The low subsurface T and $\delta^{18}O_{w-ivc}$ during the Early Holocene can be explained with an intrusion of transitional Eastern North Atlantic Water (ENAW) reaching the Azores coring site (Figure 5a). This intrusion is probably related to the influence of meltwater from the Laurentide ice sheet that is also inferred from a freshening in the subpolar

North Atlantic (Came et al., 2007; Farmer et al., 2011; Thornalley et al., 2009), although the $\delta^{18}O_{\text{w-ivc}}$ needs to be confirmed by independent proxies.

During the Mid Holocene transitional phase (8-6 ka BP), subtropical and subpolar subsurface records diverge. The subtropical $T_{\text{sub}}$ and subsurface $\delta^{18}O_{\text{w-ivc}}$ values start to converge towards those from the subtropical Blake Outer Ridge record after 8.2 ka BP, while the subpolar $T_{\text{sub}}$ record remains in the Early Holocene mode and the $\delta^{18}O_{\text{w-ivc}}$ values imply further freshening. This state coincides with a northward migration of the AF. Warm subsurface waters reached the Azores coring site, but were not transported further northward yet (Figure 5b).

At 7 ka BP subtropical subsurface records from the Azores (GEOFAR KF16) reach values similar to those of the western subtropical record from Blake Outer Ridge (ODP1058). The subpolar records (RAPID 12-1) are following the changes in the subtropical Azores records with a time of least 1,000 years. This indicates the onset of modern transport of warm subsurface waters from the subtropics into the subpolar region (Figure 5c). At the same time, LSW formation started with a prominent salinity increase and in surface waters south of Greenland (Figure 4) increasing NAC water influence in the Labrador Sea (Perner et al. 2013) and decreasing warm water transport within the NAC into the Nordic Seas (Solignac et al., 2004; Staines-Uréas et al., 2013) followed by increasing salinities at Orphan Knoll (Figure 4) (Solignac et al., 2004). These changes are potentially accompanied by a divergence of the warm water transport into the Irminger Basin and Labrador Sea at subsurface. To investigate this transport route further high- resolution SST and $T_{\text{sub}}$ reconstructions from the subpolar gyre would be needed.

In the Late Holocene after 6 ka BP, the western subtropical, subtropical Azores and subpolar records are showing parallel patterns with millennial-scale fluctuations in S temperature and T salinity that are anti-correlated with records from the Labrador Sea as already described by Thornalley et al. (2009) and Cléroux et al. (2012). With this circulation mode, short-term variability including anti-phasing between the SPG and STG temperature and salinity records is observed. Our data thus are in harmony with earlier suggestions of anti-phasing between gyre properties during for the Late Holocene records (Cléroux et al., 2012; Thornalley et al., 2009), this variability may imply that short-term freshening events in the Labrador Sea can be balanced by increased warm and saline water transport between the SPG and the STG stabilizing the AMOC. However, our data indicate that the major pathway in this stabilization mechanism at the millennial time scale must be the subsurface warm water route in the North Atlantic.

Surprisingly, no changes in the long-term trend of the Late Holocene are observed after 4 ka BP when changes in the *G. ruber* w. abundance indicate a southward movement of the AF position (Repschläger et al., 2015), that can be associated with a global southward shift of the wind systems (Wanner, 2008).

### 5.2 Driving factors for subsurface transport changes and AF position

In light of observed and anticipated anthropogenic climate change it is important to understand the underlying mechanism of Holocene climate variability. In addition to the forcing by increases in atmospheric $CO_2$, four driving factors may be of

importance for the observed Holocene changes at the Azores region: 1) changes in AC transport strength and position due to changes in MOW strength, 2) changes in the atmospheric circulation related to the NH summer insolation (Renssen, 2005) and changes in NAO patterns (e.g. Olsen et al., 2012; Wassenburg et al., 2016), 3) changes in meltwater forcing over the North Atlantic region due to melting of continental ice sheets (Mayewski et al., 2004), and 4) changes in solar activity (Bond et al., 2001), the latter will not be discussed due to the relatively low resolution of the late Holocene record which hampers time series analyses. In the following we will discuss which of those factors one to three can be identified as main cause of the variability in subsurface transport.

1) According to modeling studies (Volkov and Fu, 2010) the strength of the AC is closely linked to the strength of Mediterranean Outflow Water (MOW). Nevertheless little is known about its influence on the AC position and the question arises whether observed changes in AC position can be related to changes in MOW strength. Reconstructions of MOW strength based on contourite grain size data from the Gulf of Cadiz indicate that MOW was sluggish during Early Holocene and strengthened after 8 ka BP. During the last 2 millennia MOW strength decreased again (Rogerson et al., 2012; Toucanne et al., 2007). Thus the timing of the early Holocene MOW strengthening roughly coincides with a northward movement of the AF/AC and a strengthening of the subsurface warm water transport. The Late Holocene southward movement of the AF/AC, reconstructed from G. ruber w. abundance at 4 ka BP, leads the MOW weakening by about 2 ka BP, and does not exactly match the AF/AC signal. Though a connection between more intense MOW and increasing subsurface warm water transport in the AC/AF is likely, a connection between AC position and MOW strength cannot be definitely proved. MOW outflow is mainly driven by the density gradient between the Mediterranean Sea and the Strait of Gibraltar (Ivanovic et al., 2014 and citations therein), the latter is probably governed by the strength of the thermohaline circulation in the North Atlantic as well as by atmospheric circulation changes (Bozec et al., 2011; Voelker et al., 2006). If both are also closely related to the position and strength of the AC/AF this suggests that atmospheric and thermohaline circulation act as a common driver for the Early to Mid Holocene changes in the AF/AC position and MOW strength rather than MOW strength being the main driver in the position of the AC/AF.

2) Paleoceanographic reconstructions and modelling studies suggest that the early to Mid Holocene northern hemisphere summer insolation maximum lead to a weakening of the pressure gradient between the tropical and subpolar North Atlantic. This relaxation of pressure probably led to a northward movement of the westerly wind belt (Renssen, 2005), that controls the position of the AC/AF. The exact position of the westerly wind belt is still controversially discussed, including dynamics of the North Atlantic oscillation (NAO). NAO is the most prominent modern atmospheric circulation variability in the North Atlantic region. It is defined by the pressure difference between the Icelandic high - pressure zone and the Azores low pressure cell. $NAO^{+/-}$ phases are characterized by a strong/weak pressure gradient between Iceland and the Azores and strengthening/weakening of the westerlies (Hurrell and Deser, 2009; Hurrell et al., 2001). As the latter strength is assumed to have a major influence on the AF position, NAO changes might a driver of its changes, furthermore, are NAO changes also assumed to influence the strength of the STG/SPG dynamics.

The influence of the insolation driven early Holocene thermal maximum on NAO has been controversially discussed, either leading to a more positive (Olsen et al., 2012; Wanner, 2008) or negative NAO mode (Morley et al., 2014). Modelling studies (Gladstone et al., 2005) could not confirm a Mid Holocene NAO$^{+}$ mode and a new speleotheme dataset reconstructed multi-decadal NAO$^{+/-}$ oscillations that were active over the entire Holocene (Wassenburg et al., 2016). Such a short-term variability is not resolved in our records, and cannot account for the long- term subsurface transport changes. Thus NAO patterns seem to play a subordinate role for the variability observed in our records.

Additional to multi-decadal NAO$^{+/-}$ oscillations, Wassenburg et al., (2016) reconstructed early to Mid Holocene changes in the geometry of the Icelandic Low and Azores High pressure cell that were accompanied by a redirection of the westerlies in southwesterly wind direction during early Holocene (11-9 ka BP). These changes cannot be driven by insolation changes alone, and seem to be strongly overprinted by the extent of Laurentide ice sheets and meltwater surges as shown in a modelling study using a fully coupled earth system model (Wassenburg et al., 2016).

A change in wind direction between 8 and 9 ka BP into a Mid Holocene mode with a northward movement of the westerlies are in agreement with our *G. ruber* w. data that indicate a northward movement of the AF between 8 and 9 ka BP. A southward movement of the AF is indicated in our *G. ruber* w. data and can be associated strengthening of the westerly wind belt at about 4 ka BP, the latter matching large scale reorganization of the atmospheric circulation, including a southward shift in the westerly wind belt and in the Intertropical Convergence Zone (ITCZ) position (Wanner, 2008 and citations therein).

Despite the coincidence of frontal movements with Holocene changes in the northern hemisphere wind fields, inferred changes in the AF position do not coincide with the onset of the northward subsurface water transport. The onset of the subsurface transport and LSW formation lag behind the northward movement of the AF by about 2 ka and lead the southward movement of the AF at 4 ka BP by 2 ka. Thus, the onset of the subsurface transport seems to be decoupled from the strength of the atmospheric circulation.

3) Given the strong impact of the ice-sheets and meltwater on the atmospheric circulation, as the main driver of the early to Mid-Holocene changes (Wassenburg et al., 2016), we examine the timing and changes in meltwater more closely as a potential driver for changes in the subsurface transport.

The timing of the increased subsurface transport at about 7 ka BP into the Labrador Sea coincides well with the termination of the global sea level rise at that time (Austermann et al., 2013; Wanner, 2008), the end of meltwater flow into the Labrador Sea (Jennings et al., 2015) and increasing transport of Atlantic water into the Labrador Sea (Perner et al., 2013). The onset of the northward subsurface warm water transport agrees with the stabilization of the North Atlantic deepwater circulation that previously was related to the onset of LSW formation (Hoogakker et al., 2011; Thornalley et al., 2013). Alternatively, as recently discussed (Blaschek et al., 2015) this stabilisation may be mainly related to an increase in Nordic Seas deepwater convection, though more studies are needed to confirm this theory.

## 6. Conclusion

In this study we show a stepwise evolution of the mixed layer temperature and salinity at the AF during the Holocene that is closely linked to a northward migration of the AF and related transport of warm water within the NAC system.

The mean surface warm water transport from the tropics into the subpolar North Atlantic remained relatively stable over the last 11 ka BP. In contrast, subsurface transport into the North Atlantic evolved in three phases: an early Holocene meltwater phase (11-8 ka BP) with no subsurface transport, a mid Holocene transitional phase (8-6 ka BP) with subsurface transport that reached the AF but not the subpolar NA and a late Holocene to modern phase (6-0 ka BP) with subsurface transport into the subpolar NA that coincides with the onset of LSW formation. Within theses scenarios changes in the AF position seem to be driven by changes in the atmospheric circulation and decoupled from subsurface transport changes. The latter are mainly driven by the amount of meltwater in the subpolar North Atlantic that are probably more important for the AMOC evolution than changes in the atmospheric circulation. Multi-millennial scale variability in the subsurface transport between 10 and 6 ka BP is in anti-phase with Labrador Sea records as well as between the STG and SPG subsurface water properties, similar to previous studies postulating short-term anti-phase inter-gyre dynamics as a potential stabilizing mechanism for the modern LSW formation. Accordingly, NA subsurface transport pathways appear to have had an important role in the Mid- to Late Holocene climate stabilization. However, this aspect needs to be investigated further in high latitudes in order to obtain a full understanding of underlying forcing mechanisms.

**Data availability**

Data is available under https://doi.pangaea.de/10.1594/PANGAEA.868108.

**Acknowledgments**

We acknowledge M. Regenberg and K. Bremer for laboratory assistance and M. Kienast for final proofreading. The presented data are obtained from core GEOFAR KF16, that was generously provided by Mr. Bernard Dennielou and IFREMER. The work presented here was supported by the European Science Foundation (ESF) within the EUROCORES Program EuroMARC through contract No. ERAS-CT-2003-980409 of the European Comission, DG Research, FPG. The German Science Foundation (DFG) also contributed financial support (WE 2679/6-1).

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

**Figures**

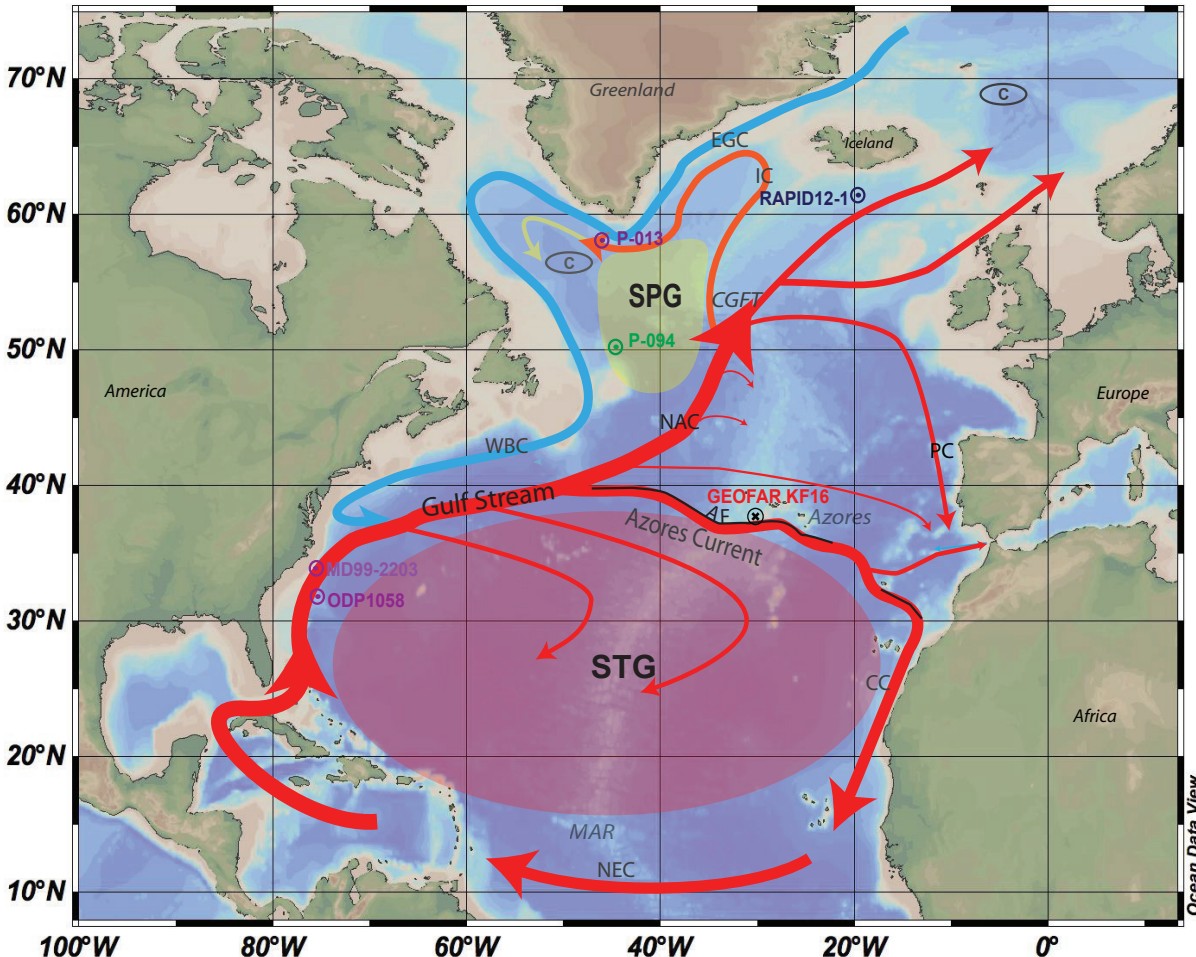

**Figure 1: Modern surface water hydrography of the North Atlantic currents are modified from (Schott et al., 2004) and (Lherminier et al., 2010), basic maps stems from ocean data view (Schlitzer, 2012) warm surface currents are shown by red arrows, black line indicates the position of the Azores fronts (Abbreviations AC Azores Current, AF Azores Front, FC Florida Current, IC Irminger Current, NAC North Atlantic Current, NEC North Equatorial Current, STG Subtropical Gyre, PC Portugal Current, TW Transitional Water, WBC Western Boundary Current), © symbols indicate the position of deepwater convection sites, the Mid Atlantic Ridge is marked with MAR.**

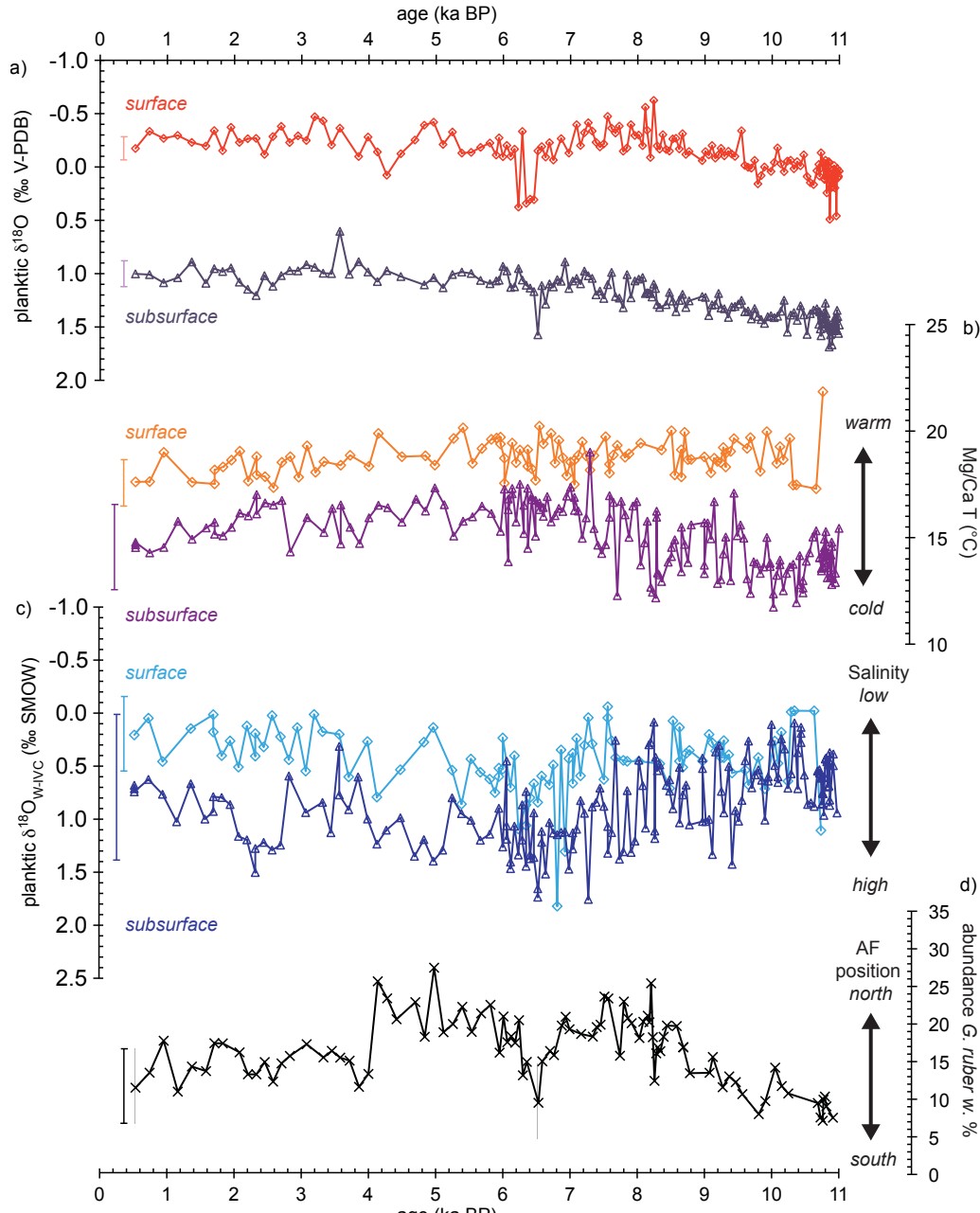

**Figure 2: Results obtained from cores GEOFAR KF16: a) δ¹⁸O records from surface (*G. ruber* w.) (red, diamonds, Schwab et al., 2012) and subsurface dwelling planktonic foraminifera (*G. truncatulinoides*) (dark blue, triangles, (Repschläger et al., 2015), b) Mg/Ca SST records from surface (*G. ruber* w.) (orange, diamonds) and subsurface (*G. truncatulinoides*) (violet, triangles), c) δ¹⁸O$_{w-ivc}$ records from surface (*G. ruber* w.) (light blue, diamonds) and subsurface (*G. truncatulinoides*) dwelling planktonic foraminifera (blue, triangles), d) *G. ruber* w. abundance (Weinelt et al., 2015) as indicator for the AF position (black crosses), grey bars indicate the 2σ std for all records.**

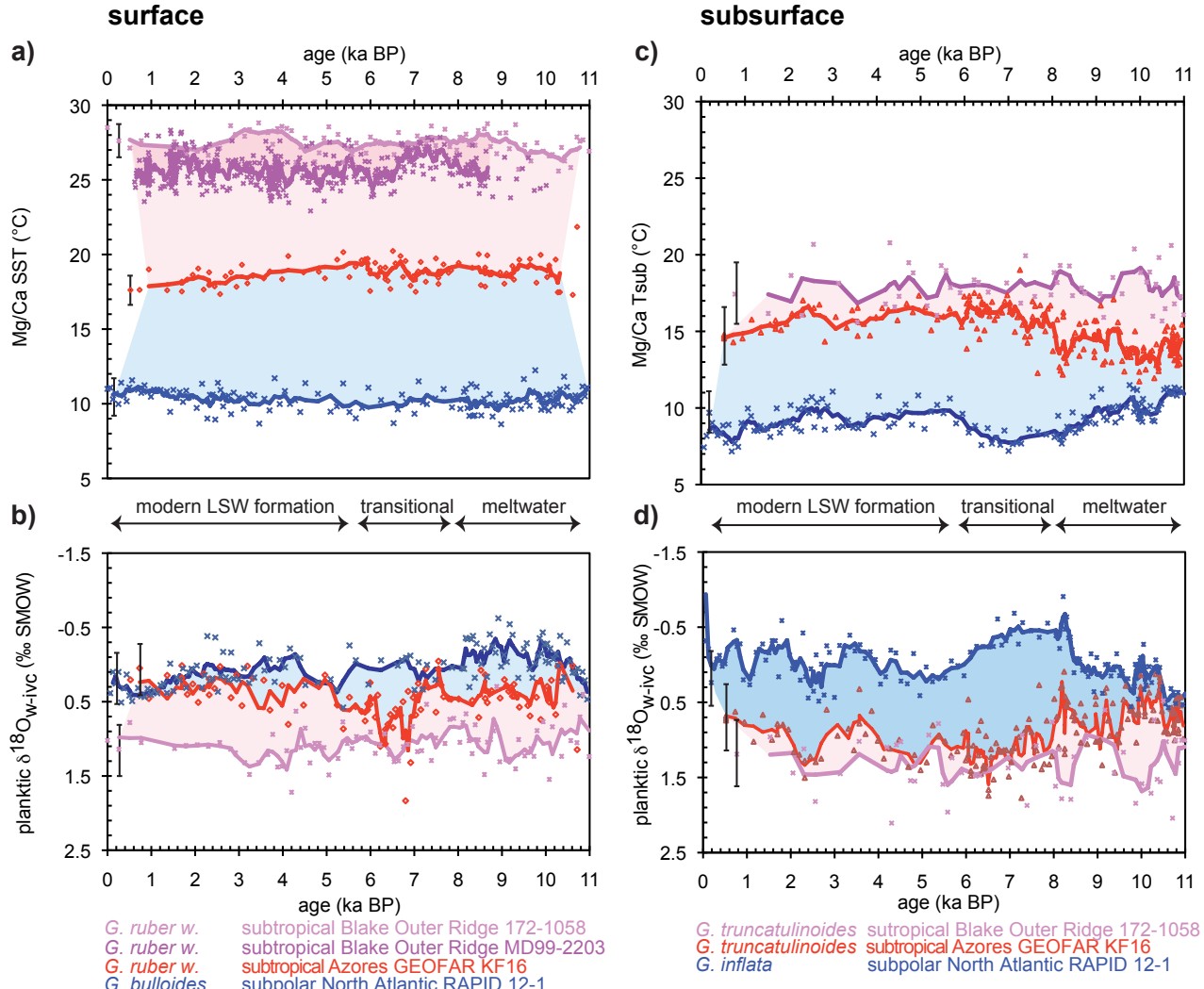

**surface**

**a)**

**subsurface**

**c)**

**b)**

modern LSW formation    transitional    meltwater

**d)**

modern LSW formation    transitional    meltwater

*G. ruber w.*        subtropical Blake Outer Ridge 172-1058
*G. ruber w.*        subtropical Blake Outer Ridge MD99-2203
*G. ruber w.*        subtropical Azores GEOFAR KF16
*G. bulloides*       subpolar North Atlantic RAPID 12-1

*G. truncatulinoides*    sutropical Blake Outer Ridge 172-1058
*G. truncatulinoides*    subtropical Azores GEOFAR KF16
*G. inflata*             subpolar North Atlantic RAPID 12-1

**Figure 3: Surface and subsurface temperature and salinity reconstructions from tropical to subpolar North Atlantic. Left panel: Comparison between a) surface temperatures and b) $\delta^{18}O_{w\text{-}ivc}$ records from the tropical (pink) (Bahr et al., 2013), subtropical (red), and subpolar (blue) (Thornalley et al., 2009) North Atlantic. Right panel: Comparison between c) subsurface temperatures and d) $\delta^{18}O_{w\text{-}ivc}$ records from the tropical (pink) (Bahr et al., 2013), subtropical (red), and subpolar (blue) (Thornalley et al., 2009), grey bars indicate the std for all records.**

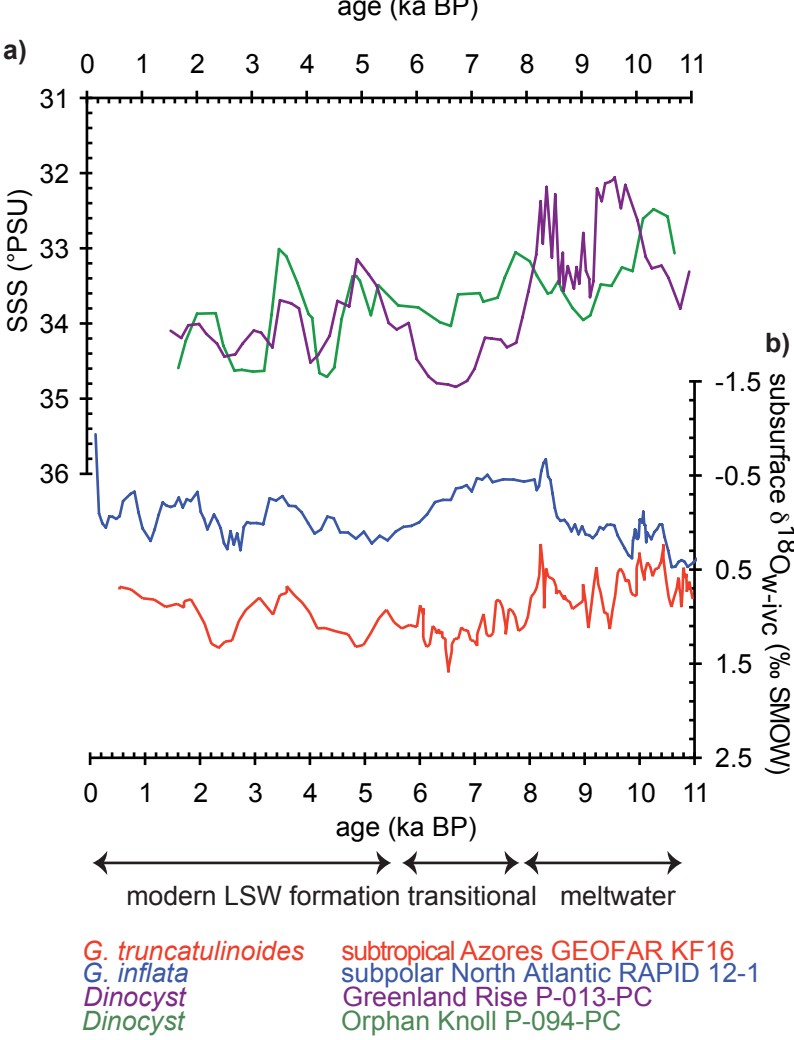

**Figure 4: Comparison between a) SSS reconstructions (summer season estimates) based on dinoflagellate cyst assemblages from the northern and southern entrances of the Labrador Sea (Solignac et al., 2004) and b) Subsurface salinity reconstructions from subtropical to subpolar North Atlantic δ$^{18}$O$_{w-ivc}$ records, subtropical (red), and subpolar (blue) (Thornalley et al., 2009).**

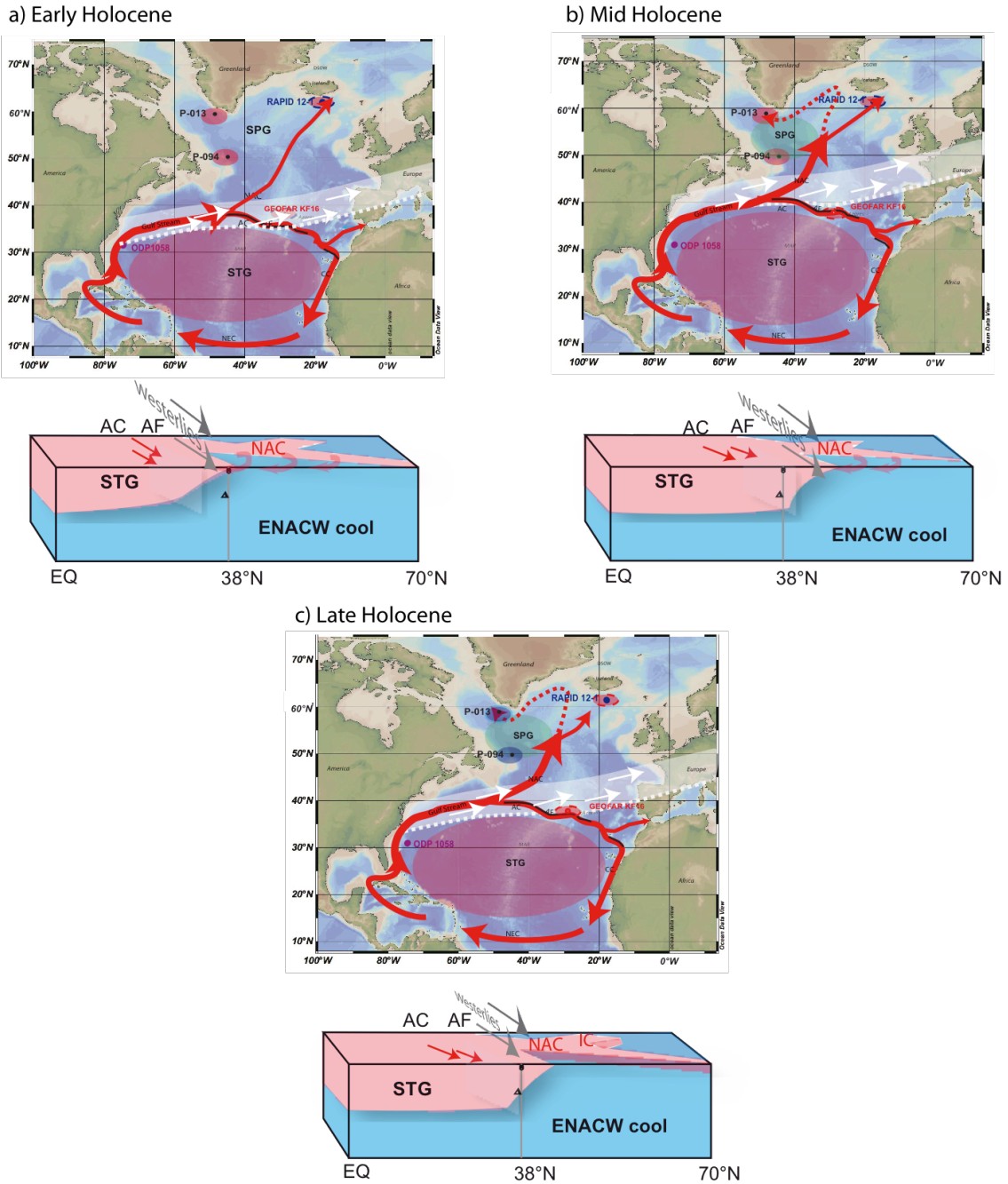

**Figure 5: Conceptual model showing the three different states of subsurface transport a) early Holocene meltwater phase, b) mid Holocene transitional phase, and c) late Holocene modern phase. Abbreviations: AC Azores Current, AF Azores Front, ENACW Eastern North Atlantic Central Water, NAC North Atlantic Current, STG Subtropical Gyre.**

