# Peer review of "Holocene evolution of the North Atlantic subsurface transport"

_Climate of the Past, 2016_

## Referee Comment (RC1) · Anonymous Referee #1 · 19 Dec 2016

The authors investigate the dynamics of the subtropical vs. subpolar gyres in the North Atlantic focusing on the transport pathways between both circulation systems. They find that subsurface transport plays a so far underestimated role for the heat and salt exchange between high and low latitudes. It is inferred that freshwater forcing is the dominant process governing the reconstructed (sub)surface temperature and salinity changes. The distinct subsurface variability is an important outcome of the study and sheds new light on the dynamics of the subtropical and subpolar gyre interaction. As such the paper merits publication, however, there are some issues that should be considered before final publication:

General comments:

1) Wind forcing is clearly an important driver of subtropical gyre dynamics, in particular for the location of the Azores Front/Current (AF/AC). However, modelling studies suggest that the Mediterranean Outflow Water (MOW) exerts a pivotal influence as well, with indications that there would be no AC without MOW (Özgökmen et al., 2001; Volkov and Fu, 2010). MOW strength varies significantly during the Holocene with a weak outflow during the early Holocene (e.g. (Rogerson et al., 2012)) – how does this affect the position of the AC/AF? Could the northward shift of the AF indicated by increased G. ruber abundances until 4 ka might be at least in parts also due to a less strong MOW giving rise to a more northern location of the AF? Although I agree that wind-fields are most important for the positioning of the AF, the authors should at least briefly discuss the potential influence of the MOW.

2) The authors discuss at length the inverse relation of the STG variability to paleoenvironmental reconstructions from the Labrador Sea without showing any record from Labrador Sea such as P-012 and p-094 from Fig. 4. Given the importance of this point for the entire reasoning within the manuscript the discussed records from the Labrador Sea must be included in the figures.

3) It is not stated whether the right or left-coiling variety of G. truncatulinoides was picked or whether both were pooled. Studies clearly show that both morphotypes have different temperature preferences (Billups et al., 2016; Thiede, 1971), hence, it is therefore important not to pool right and left coiling G. truncatulinoides. Furthermore, in the methods it should be stated from which size fractions the foraminifera (both, G. ruber and G. truncatulinoides) were picked. Particularly Mg/Ca shows distinct ontogenetic effects and size-dependent offsets (Elderfield et al., 2002; Friedrich et al., 2012).

Specific comments:

ABSTRACT Line 2, "... interactions with the subtropical gyre..." is quite unspecific. Maybe replace by "... advection of saline water from the subtropical gyre". I. 11, "published data": specify which type of data (salinity and temperature). I. 13/14, "Subsurface warm water transport started at about 8 ka with subtropical heat storage,..." this
sentence is again quite unspecific. Specify: What is the direction of the heat transport and what is the exact relation to heat storage?

INTRODUCTION p. 1, l. 21: remove comma after "due to" p. 1, l. 30: replace "ocean" with "oceanic" p.2., l. 29: "entertained" is an odd phrasing here. Better replace with e.g. "discussed".

REGIONAL SETTING p. 3, I. 10 (and elsewhere): "G. ruber w.": "w." should not be in italics.

METHODS This section is quite superficial regarding the analytical details: - please state the number of individual forams picked for stable isotope and trace element analyses. - were Mg/Ca measurements monitored for contamination by checking e.g. Al, Fe, Mn vs. Ca ratios? - How were the samples cleaned? With or without reductive cleaning? These are guite elementary analytical information! - Which machine was used for  $\delta$ 18O and Mg/Ca analysis at which laboratory? - It would be good to have a brief reference to the age model. p. 3, l. 24 - : please state the equations used for Mg/Ca – temperature conversion explicitly. p. 3, l. 26: "(Repschläger et al., 2015)" please check for the format of the citation (also in other parts of the text). p. 4, I. 7: there are two ".." after "comparison". p. 4, I. 9: instead of using the low-resolution sea level correction after Waelbroeck et al. (2002), a more recent sea level curve should be used such as Austermann et al. (2013) p. 4, l. 9: check format of "Cleroux 2011" p. 4., l. 10: explicitly state the equation (S8) from Cleroux et al. (2011) here. p. 4, l. 11: "5- point": remove blank p. 4, I. 13: Weinelt et al. (2015) is missing in the reference list. general: please introduce "w-ivc" as abbreviation. Note that the respective x-axis in Fig. 2C is labeled with "SW-IV".

RESULTS p. 4, l. 21: add ",respectively" after "8 ka BP". p. 4, l. 22-23: the positive excursion in  $\delta$ 18O: might there be a relation to the discontinuity mentioned later in the text? p. 4, l. 24 (and elsewhere): check for consistency that there is no blank before "°C" p. 4, l. 29: "single points cannot be interpreted": this sentence is somewhat
nonsensical as single points should be not interpreted in general. p. 4, l. 29: "indicate a evolution": insert a reference to Fig. 3 here. p. 4, l. 30: there seems to be a number missing after "10.5 and" p. 5, l. 1: there seems to be an "and" missing in "1.5 ‰ 1.7‰1 p. 5, l. 10: insert an "in" after "any changes"

DISCUSSION p. 5, I. 12-13: as mentioned for the introduction there is no need to introduce the INTIMATE stratigraphy here. p. 5, I. 25 (and Fig. 3): site ODP 1058 and core MD99-2203 are from subtropical waters, not tropical sites. Both are also not from the Caribbean as stated in the Fig. 3 caption. Better refer to western subtropical Atlantic. p. 6, I. 3: delete comma after "above" p. 6, I. 5: insert "." after "trends" p. 6, I. 7: "very similar" is an overstatement in my opinion p. 6, I. 8: not sure what is meant by "12/10°C". Does this refer to the temperature range of Tsub? p. 7, I. 4: insert a blank before "(Repschläger et al., 2015)" p. 7, I. 29-31: this paragraph is too speculative and should be omitted, unless the authors prove the existence of the 1500 yrs-cycle by spectral analysis. p. 8, I. 10: the reference to "AMOC evolution" is somewhat misleading in this context as the authors reconstruct in the first place the dynamics of the STG and SPG.

FIGURES Fig. 1: Indicate the position of core MD99-2203. Please place the circle indicating the position of ODP 1058 above the red arrow. The blue arrow is at places hard to see, a darker tone would help here. Fig. 2: In the caption a reference is missing to the source of the G. ruber abundance data (Weinelt et al., 2015). The readability of the caption would benefit if the listing of items is separated by comma (e.g. "..., c)..." and a full stop at the end. In general, the error bars at individual points suggest that they represent individual replicates which is not the case. It would be better to state the 2sigma-error next to the respective y-axis. Fig. 3: Please put "18"in  $\delta$ 180 into superscript. Fig. 4: The listing of items in the caption should be separated by comma (e.g. "..., c)..."

References: Austermann, J., Mitrovica, J. X., Latychev, K., et al., 2013, Barbadosbased estimate of ice volume at Last Glacial Maximum affected by subducted plate:

**CPD**
Nature Geosci, 6, 553-557. Billups, K., Hudson, C., Kunz, H., et al., 2016, Exploring Globorotalia truncatulinoides coiling ratios as a proxy for subtropical gyre dynamics in the northwestern Atlantic Ocean during late Pleistocene Ice Ages: Paleoceanography. Elderfield, H., Vautravers, M. J., and Cooper, M., 2002, The relationship between shell size and Mg/Ca, Sr/Ca, d18O, and d13C of species of planktonic foraminifera: Geochem. Geophys. Geosyst., 3. Friedrich, O., Schiebel, R., Wilson, P. A., et al., 2012, Influence of test size, water depth, and ecology on Mg/Ca, Sr/Ca,  $\delta$ 18O and  $\delta$ 13C in nine modern species of planktic foraminifers: Earth and Planetary Science Letters, 319-320, 133-145. Özgökmen, T. M., Chassignet, E. P., and Rooth, C. G., 2001. On the connection between the Mediterranean outflow and the Azores Current: Journal of Physical Oceanography, 31, 461-480. Rogerson, M., Rohling, E. J., Bigg, G. R., et al., 2012, Paleoceanography of the Atlantic-Mediterranean exchange: Overview and first quantitative assessment of climatic forcing: Rev. Geophys., 50, RG2003. Thiede, J., 1971, Variations in coiling ratios of Holocene planktonic foraminifera: Deep Sea Research and Oceanographic Abstracts, 18, 823-831. Volkov, D. L., and Fu, L.-L., 2010, On the reasons for the formation and variability of the Azores Current: Journal of Physical Oceanography, 40, 2197-2220.

---

## Referee Comment (RC2) · Anonymous Referee #2 · 9 Jan 2017

General and specific comments: The paper "Holocene evolution of the North Atlantic subsurface transport" by Repschläger et al presents new Holocene surface temperature and salinity estimations from a sediment core at the Azores Front. The data are used together with existing subsurface and surface data from the same core (Repschläger et al, 2015. Paleoceanography) in order to investigate the subsurface transport between the subtropical and polar North Atlantic and the intergyre transport pathways. This is a very timely and welcome study, and it is very much within the scope of CP. Overall the paper is well structured with clear and good figures. However, it has some issues. The authors suggest a freshwater control of the subsurface transport. The discussion seems to be focused on this hypothesis, and other driving factors appear to be somewhat superficially discussed. Hence, freshwater control does not seem convincing. All driving factors need to be discussed in much more detail including additional studies, e.g. Olsen et al. (2012) DOI 10.1038/ngeo1589 in addition to model runs e.g. Blaschek et al. (2015), DOI 10.1007/s00382-014-2279-1. It should also be taken into consideration that the current reconstruction is compared to two other reconstructions. Is it possible to go even further south/north or east/west? The current study also heavily refer to the existing paper by Repschläger et al. (2015) in Paleoceanography throughout the paper. Naturally, some things are not necessary to describe in detail twice, but it should be possible to follow the current study without needing the other paper next to you. Additionally, some references to Repschläger et al. 2015 are in some places misleading, please use original references on e.g. the preferred depth habitats of the planktic foraminifera.

Technical corrections: Abstract, page 1, lines 7-15: The abstract does not mention which type of data that have been used for the reconstructions (Mg/Ca and d18O data).

Regional Setting, page 3, lines 8-12: This part belongs to discussion or introduction.

Methods, page 4, lines 1-5: Explain in more detail

Results, page 5, lines 3-8: Interpretations that belong to discussion

Discussion, page 5, line 26: Consider to add core ID in order to facilitate reading of figure.

Discussion, page 6, line 6: The reference is to Figure 3b, but it is in fact Figure 3c?

Discussion, page 6, lines 20-6: Unclear; explain in more detail.

Discussion, page 7, lines 6-31: Include additional studies and discuss all drivers in more detail

Discussion, page 7, line 15: The acronym ITCZ needs to be defined.

References, page 8, line 27: Make sure that "K0.., N" reads "Koç, N" or "Koc, N" in final version.

Figure 1: Add positions and core ID for the studies from Labrador Sea.

Figure 3: In the figure caption data are described and interpreted; this should be removed.

Figure 4: The acronyms on the figure should be defined in the figure text in order to facilitate reading.

СЗ

---

## Author Comment (AC1) · 17 Feb 2017

Response to the reviewers on behalf of all co-authors

Response to Referee #1

We greatly acknowledge the thoughtful scientific comments and very detailed technical hints for corrections in the text and figures of Referee #1. They will improve the manuscript considerably.

Although in general positive towards our study, Referee #1 asked for a critical reconsideration of three major issues:

1) Referee #1 proposed to include a brief discussion about the role of the strength of Mediterranean outflow water (MOW) on the position of the Azores Front / Azores

Current (AF/AC), as MOW may exert a pivotal influence on the existence of the AC (Volkov et Fu 2010). For example, it should be discussed whether the apparent Mid Holocene northward movement of the AF/AC might be related to a weakening of the MOW is likely or not.

To our opinion, according to modeling studies (e.g. Volkov and Fu, 2010) MOW is indeed closely linked to the strength of the AC, however, little is known about its influence on the AC position. Reconstructions of MOW strength based on contourite grain size data from the Gulf of Cadiz indicate that MOW was sluggish during Early Holocene and strengthened after 8 ka BP. During the last 2 millennia MOW strength decreased again (Rogerson et al., 2012; Toucanne et al., 2007). Thus the timing of the early Holocene MOW strengthening roughly coincides with an increased subsurface warm water transport towards the Azores. On the other hand, the Late Holocene southward movement of the AF/AC, reconstructed from G. ruber w. abundance at 4 ka BP, corresponds with MOW weakening although the timing of the latter at about 2 ka BP, does not exactly match the AF/AC signal. Thus a connection between more intense MOW and increasing subsurface warm water transport in the AC/AF seems likely. However, the coincidence of a Mid-Holocene increase in MOW strength with a more northerly AF/AC position opposes the suggestion of Referee #1 that a sluggish MOW would lead to a northward movement of the AF. Since MOW outflow is mainly driven by the density gradient between the Mediterranean Sea and the Strait of Gibraltar (Ivanovic et al., 2014 and citations therein), the latter is probably governed by the strength of the thermohaline circulation in the North Atlantic as well as by atmospheric circulation changes (Bozec et al., 2011; Voelker et al., 2006). If both are also closely related to the position and strength of the AC/AF this suggests that atmospheric and thermohaline circulation act as a common driver for the Early to Mid Holocene changes in the AF/AC position and MOW strength rather than MOW strength being the main driver in the position of the AC/AF. As suggested by Referee #1 we will include this paragraph discussing the potential role of MOW for the AF/AC position ...

2) Referee #1 proposed to include the Labrador Sea temperature and salinity records from P-012 and P-094 (Solignac et al., 2004) in one of our figures. We agree that the reasoning in the manuscript will strongly benefit from including theses records and will do so in a revised version.

3) Referee #1 asked to specify size fractions of G. ruber w. and G. truncatulinoides as well as the coiling direction of G. truncatulinoides of the specimens used for analyses in order to avoid species specific and size related offsets in temperature reconstructions. Furthermore, under specific comments referee #1 asked to improve the methods section by providing more specific details. In order to address all methods related questions, we suggest to rewrite the methods chapter for a revised manuscript version as follows: "For our study we used core GEOFAR KF16 taken at 38°N, south of the Azores Islands at the Mid Atlantic Ridge, from 3060 m water depth. This position in the vicinity of the AF is ideal to trace changes in STG position and associated varying influence of subtropical and temperate waters (Repschläger et al., 2015). The age model for core GEOFAR KF16 is published in Schwab et al. (2012). Results are compared to published data from the subtropical (Bahr et al., 2013; Cléroux et al., 2012) and subpolar North Atlantic (Thornalley et al., 2009) along the warm water route. We combine published  $\delta$ 18O data of planktonic surface and subsurface dwelling for aminifera G. ruber w. (Schwab et al., 2012) and G. truncatulinoides (Repschläger et al., 2015) with new Mg/Ca records from the same species. All measurements were conducted on samples of the size fraction 315-400  $\mu$ m on monospecific samples of 10-25 specimens of G. ruber w. and 15 specimens of mixed left and right coiling G. truncatulinoides. Stable isotope analyses were carried out at Leibniz Laboratory for Radiometric Dating and Stable Isotope Research at Kiel University. For analyses a Finnigan MAT 523 mass spectrometer coupled with a Kiel I carbon preparation device was used and results were calibrated to the Vienna Pee Dee Belemnite (V-PDB) scale. The 2  $\sigma$  standard deviation (std) obtained from 10 replicates of downcore samples was +/- 0.11 ‰ for G. ruber w. and +/- 0.12 ‰ for G. truncatulinoides. Mg/Ca analyses followed the cleaning procedure of Martin and Lea (2002) and Repschläger et al. (2015), using reductive

and oxidative cleaning. Measurements were carried out with a simultaneous inductively coupled plasma-optical emission spectrometry (ICP-OES) instrument with radial plasma observation. Potential shell contamination or coatings by authigenic minerals were monitored using additional Fe, AI and Mn measurements. Analytical 2  $\sigma$  std was +/-0.20 mmol/mol Mg/Ca for G. ruber w. and +/-0.14 mmol/mol Mg/Ca for G. truncatulinoides. To convert Mg/Ca values into water temperature estimates, we used the principle equation format Mg/Ca = b exp ( $a^{T}$ ) with species-specific variables a=0.76 and b=0.07 for G. ruber w. and a=0.78 and b=0.04 for G. truncatulinoides (equation for mixed subsurface dwellers) published by Cleroux et al. (2008). The summed analytical and calibration 2  $\sigma$  std is +/- 1°C for G. ruber w. and +/- 2°C for G. truncatulinoides. The calibrated Mg/Ca temperature estimates for both species in our core top samples match well with modern surface and subsurface (200 m depth) temperatures at the Azores coring site. Seasonal temperature effects on the foraminiferal Mg/Ca signal are assumed to play a subordinate role at our coring site (Cléroux et al., 2008; Repschläger et al., 2015). In addition, we assume that the subsurface temperature signal of G. truncatulinoides is predominantly determined by the conditions at the AF and not by the migration of G. truncatulinoides to shallower, warmer water depths or by thermocline shoaling (see supplementary information and Repschläger et al. (2015)). Changes in salinity are reconstructed following the procedure described in Repschläger et al. (2015). The temperature effect, estimated from the Mg/Ca was removed from the for a miniferal  $\delta$ 180 carbonate using the general equation of Shackleton (1974) for both, G. ruber w. and G. truncatulinoides, in order to be consistent with the datasets used for comparison. A correction for VPDB to SMOW was included resulting seawater  $\delta$ 180w values discussed in the following. For estimation of  $\delta$ 180w uncertainties we followed the approach of Cleroux et al., (2011) and used their equation S8:

 $\sigma$  ( $\delta$ 18Ow-ivc) = (( $\sigma$  ( $\delta$ 18Oforam))2+( $\sigma$  (Temp))2\*(0.23)2)1/2

leading to calculated 2  $\sigma$  std of +/- 0.35 ‰ and +/- 0.68 ‰ for G. ruber w and G. truncatulinoides, respectively. Given the relative large error in temperature and  $\delta$ 180w reconstructions, we only used 5-point average time series for interpretation to investigate longer-term trends in the datasets. The  $\delta$ 18Ow records were corrected for ice volume, using the eustatic sea level curve of Austermann et al. (2013) and are expressed as  $\delta$ 18Ow-ivc values throughout the text. Changes in the AF front were reconstructed using the relative abundance of G. ruber w. published by Weinelt et al. (2015). Abundance counts possess a 2  $\sigma$  std of +/- 2.5 %. As G. ruber w. is most abundant within the STG (Ottens, 1991; Schiebel et al., 2002; Storz et al., 2009), low/high abundances indicate a southward/northward movement of the AF relative to the coring site. Because the position of the AF is mainly related to changes in the westerly wind belt, the abundances of G. ruber w. indicate the relative contribution of subtropical water and can be used as tracer for the position of the northern STG rim and thus of the position of westerlies as discussed in Repschläger et al. (2015) before. "

Considering the concerns of Referee #1 about the used size fractions and coiling varieties of G. truncatulinoides we would like to respond referring to Friedrich et al (2012):

Despite the more recent results presented by Billups et al. (2016), the dataset of Friedrich et al. (2012) shows that G. truncatulinoides left and right coiling have similar Mg/Ca ratios in the size fraction 315-400 $\mu$ m, though only based on one measurement. Therefore we used a limited size fraction window with relative small size-dependent changes.

With respect to coiling varieties the Mg/Ca calibration dataset of Cléroux et al. (2008) shows that the difference in Mg/Ca temperature estimates between the left and right coiling variety of G. truncatulinoides is within the calibration error of +/- 2 °C that is assumed for the mixed calibration used in our work. However, we will include a sentence mentioning that pooling of G. truncatulinoides coiling varieties should be avoided to decrease the uncertainty level in Mg/Ca derived surface ocean temperatures.

Additional to the changes proposed above we will carefully revise the manuscript according to the numerous technical comments that will definitely increase the quality of

the manuscript.

The following references will be included: Austermann, J., Mitrovica, J. X., Latychev, K., and Milne, G. A.: Barbados-based estimate of ice volume at Last Glacial Maximum affected by subducted plate, Nature Geosci, 6, 553-557, 2013.

Bahr, A., Nürnberg, D., Karas, C., and Grützner, J.: Millennial-scale versus long-term dynamics in the surface and subsurface of the western North Atlantic Subtropical Gyre during Marine Isotope Stage 5, Global and Planetary Change, 111, 77-87, 2013.

Billups, K., Hudson, C., Kunz, H., and Rew, I.: Exploring Globorotalia truncatulinoides coiling ratios as a proxy for subtropical gyre dynamics in the northwestern Atlantic Ocean during late Pleistocene Ice Ages, Paleoceanography, 31, 2016PA002927, 2016.

Bozec, A., Lozier, M. S., Chassignet, E. P., and Halliwell, G. R.: On the variability of the Mediterranean Outflow Water in the North Atlantic from 1948 to 2006, J. Geophys. Res., 116, C09033, 2011.

Cléroux, C., Cortijo, E., Anand, P., Labeyrie, L., Bassinot, F., Caillon, N., and Duplessy, J.-C.: Mg/Ca and Sr/Ca ratios in planktonic foraminifera: Proxies for upper water column temperature reconstruction, Paleoceanography, 23, PA3214, 2008.

Cléroux, C., Debret, M., Cortijo, E., Duplessy, J.-C., Dewilde, F., Reijmer, J., and Massei, N.: High-resolution sea surface reconstructions off Cape Hatteras over the last 10 ka, Paleoceanography, 27, PA1205, 2012.

Friedrich, O., Schiebel, R., Wilson, P. A., Weldeab, S., Beer, C. J., Cooper, M. J., and Fiebig, J.: Influence of test size, water depth, and ecology on Mg/Ca, Sr/Ca, d18O and d13C in nine modern species of planktic foraminifers, Earth and Planetary Science Letters, 319/320, 133-145, 2012.

Ivanovic, R. F., Valdes, P. J., Gregoire, L., Flecker, R., and Gutjahr, M.: Sensitivity of modern climate to the presence, strength and salinity of Mediterranean-Atlantic exchange in a global general circulation model, Climate Dynamics, 42, 859-877, 2014.

Martin, P. A. and Lea, D. W.: A simple evaluation of cleaning procedures on fossil benthic foraminiferal Mg/Ca, Geochemistry, Geophysics, Geosystems, 3, 8401, 2002.

Ottens, J. J.: Planktic foraminifera as North Atlantic water mass indicators, Oceanologica Acta, 14, 123-140, 1991.

Repschläger, J., Weinelt, M., Kinkel, H., Andersen, N., Garbe-Schönberg, D., and Schwab, C.: Response of the subtropical North Atlantic surface hydrography on deglacial and Holocene AMOC changes, Paleoceanography, 30, 2015.

Rogerson, M., Rohling, E. J., Bigg, G. R., and Ramirez, J.: Paleoceanography of the Atlantic-Mediterranean exchange: Overview and first quantitative assessment of climatic forcing, Reviews of Geophysics, 50, RG2003, 2012.

Schiebel, R., Schmuker, B., Alves, M., and Hemleben, C.: Tracking the Recent and late Pleistocene Azores front by the distribution of planktic foraminifers, Journal of Marine Systems, 37, 213-227, 2002.

Schwab, C., Kinkel, H., Weinelt, M., and Repschläger, J.: Coccolithophore paleoproductivity and ecology response to deglacial and Holocene changes in the Azores Current System, Paleoceanography, 27, PA3210, 2012.

Shackleton, N. J.: Attainment of isotopic equilibrium ocean water and the benthonic foraminifera genus Uvigerina: Isotopic changes in the ocean during the last glacial, Cent. Natl.Rech. Sci. Collog. Int., 219, 203–209, 1974.

Solignac, S., de Vernal, A., and Hillaire-Marcel, C.: Holocene sea-surface conditions in the North Atlantic contrasted trends and regimes in the western and eastern sectors (Labrador Sea vs. Iceland Basin), Quaternary Science Reviews, 23, 319-334, 2004.

Storz, D., Schulz, H., Waniek, J. J., Schulz-Bull, D. E., and Kuçera, M.: Seasonal and interannual variability of the planktic foraminiferal flux in the vicinity of the Azores Current, Deep Sea Research Part I: Oceanographic Research Papers, 56, 107-124, 2009.

Thornalley, D. J. R., Elderfield, H., and McCave, I. N.: Holocene oscillations in temperature and salinity of the surface subpolar North Atlantic, Nature, 457, 711-714, 2009.

Toucanne, S., Mulder, T., Schönfeld, J., Hanquiez, V., Gonthier, E., Duprat, J., Cremer, M., and Zaragosi, S.: Contourites of the Gulf of Cadiz: A high-resolution record of the paleocirculation of the Mediterranean outflow water during the last 50,000 years, Palaeogeography, Palaeoclimatology, Palaeoecology, 246, 354-366, 2007.

Voelker, A. H. L., Lebreiro, S. M., Schönfeld, J., Cacho, I., Erlenkeuser, H., and Abrantes, F.: Mediterranean outflow strengthening during northern hemisphere coolings: A salt source for the glacial Atlantic?, Earth and Planetary Science Letters, 245, 39-55, 2006.

Volkov, D. L. and Fu, L.-L.: On the Reasons for the Formation and Variability of the Azores Current, Journal of Physical Oceanography, 40, 2197-2220, 2010.

Weinelt, M., Schwab, C., Kneisel, J., Hinz, M., and Climate and societal change in the western Mediterranean area around 4.2 ka BP. In: 2200 BC – Ein Klimasturz als Ursache für den Zerfall der Alten Welt? 2200 BC – A climatic breakdown as a cause for the collapse of the old world? 7. Mitteldeutscher Archäologentag vom 17. bis 19. Oktober 2013 in Halle (Saale). 7th Archaeological Conference of Central Germany October 23–26, 2014 in Halle (Saale). Meller, H., Arz, H. W., Jung, R., and Risch, R. (Eds.), Tagungen des Landesmuseums für Vorgeschichte Halle 12 Halle [Saale] 2015.

---

## Author Comment (AC3) · 17 Feb 2017

Response to the reviewer on behalf of all co-authors

Response to Referee #2

We acknowledge the comments made by Referee #2 on our manuscript. While generally positive about the results presented Referee #2 criticizes that the freshwater influence mainly discussed as controlling factor for subtropical subsurface transport is too limited and that also other mechanisms should be addressed as driving factors for Holocene changes in the inter-gyre transport.

We propose to revise the discussion by including a more detailed paragraph describing changes in the atmospheric circulation mainly referring to changes in NAO patterns

and its influence on the STG/SPG system (e.g. Olsen et al., 2012). Following the suggestion of Referee #1, also changes in MOW strength and its potential influence on the AF/AC position will be discussed. Finally, more studies (e.g.Blaschek et al., 2015) about the influence of early Holocene meltwater pathways will be included into the discussion.

Furthermore, Referee #2 proposed to extent the reconstruction of the warm water transport route with cores further south/north or east/west. Unfortunately, high-resolution Holocene studies investigating changes along the warm water route including surface and subsurface temperature and salinity reconstructions are limited. (Farmer et al., 2011) published Holocene surface and subsurface temperature and salinity time series from core MD99-2251 positioned at the eastern flank of Reykjanes Ridge underneath the inflow pathway of the NAC towards core RAPID 12-1 K. The dataset of core MD99-2251 shows the same long-term evolution over early to Mid Holocene evolution as the dataset of (Thornalley et al., 2009) and we thus will mention it in a revised manuscript. In order to highlight the connection to the Labrador Sea, a comparison between our time series and surface water datasets of (Solignac et al., 2004) will be added to one of our figures as proposed in our response to Referee #2. However, the sparse number of high-resolution subsurface temperature and $\delta$18Ow reconstructions emphasizes the need of further studies and indicates the novelty of our study.

A further aspect raised by Referee #2 was the strongly shortened methods section in the manuscript, that heavily relied on a previous publication (Repschläger et al., 2015). Indeed the methods section has been shortened very strictly in order to avoid self-plagiarism. However, we agree that it should be possible to follow the new manuscript independently. Thus the missing information will be provided in a revised version of the paper using a revised paragraph also requested by Referee #1:

"For our study we used core GEOFAR KF16 taken at 38°N, south of the Azores Islands at the Mid Atlantic Ridge, from 3060 m water depth. This position in the vicinity

of the AF is ideal to trace changes in STG position and associated varying influence of subtropical and temperate waters (Repschläger et al., 2015). The age model for core GEOFAR KF16 is published in (Schwab et al., 2012). Results are compared to published data from the subtropical (Bahr et al., 2013; Cleroux et al., 2012) and subpolar North Atlantic (Thornalley et al., 2009) along the warm water route. We combine published $\delta$18O data of planktonic surface and subsurface dwelling foraminifera G. ruber w. (Schwab et al., 2012) and G. truncatulinoides (Repschläger et al., 2015) with new Mg/Ca records from the same species. All measurements were conducted on samples of the size fraction 315-400 $\mu$m on monospecific samples of 10-25 specimens of G. ruber w. and 15 specimens of mixed left and right coiling G. truncatulinoides. Stable isotope analyses were carried out at Leibniz Laboratory for Radiometric Dating and Stable Isotope Research at Kiel University. For analyses a Finnigan MAT 523 mass spectrometer coupled with a Kiel I carbon preparation device was used and results were calibrated to the Vienna Pee Dee Belemnite (V-PDB) scale. The 2 $\sigma$ standard deviation (std) obtained from 10 replicates of downcore samples was +/- 0.11 ‰ for G. ruber w. and +/- 0.12 ‰ for G. truncatulinoides. Mg/Ca analyses followed the cleaning procedure of (Martin and Lea, 2002; Repschläger et al., 2015), using reductive and oxidative cleaning. Measurements were carried out with a simultaneous inductively coupled plasma-optical emission spectrometry (ICP-OES) instrument with radial plasma observation. Potential shell contamination or coatings by authigenic minerals were monitored using additional Fe, Al and Mn measurements. Analytical 2 $\sigma$ std was +/-0.20 mmol/mol Mg/Ca for G. ruber w. and +/-0.14 mmol/mol Mg/Ca for G. truncatulinoides. To convert Mg/Ca values into water temperature estimates, we used the principle equation format Mg/Ca = b exp (a*T) with species-specific variables a=0.76 and b=0.07 for G. ruber w. and a=0.78 and b=0.04 for G. truncatulinoides (equation for mixed subsurface dwellers) published by (Cleroux et al. 2008). The summed analytical and calibration 2 $\sigma$ std is +/- 1°C for G. ruber w. and +/- 2°C for G. truncatulinoides. The calibrated Mg/Ca temperature estimates for both species in our core top samples match well with modern surface and subsurface (200 m depth) temperatures at the

Azores coring site. Seasonal temperature effects on the foraminiferal Mg/Ca signal are assumed to play a subordinate role at our coring site (Cleroux et al., 2008; Repschläger et al., 2015). In addition, we assume that the subsurface temperature signal of G. truncatulinoides is predominantly determined by the conditions at the AF and not by the migration of G. truncatulinoides to shallower, warmer water depths or by thermocline shoaling (see supplementary information and (Repschläger et al., 2015)). Changes in salinity are reconstructed following the procedure described in Repschläger et al. (2015). The temperature effect, estimated from the Mg/Ca was removed from the foraminiferal $\delta$18Ocarbonate using the general equation of Shackleton (1974) for both, G. ruber w. and G. truncatulinoides, in order to be consistent with the datasets used for comparison. A correction for VPDB to SMOW was included resulting seawater $\delta$18Ow values discussed in the following. For estimation of $\delta$18Ow uncertainties we followed the approach of Cleroux et al., (2011) and used their equation S8:

$\sigma$ ($\delta$18Ow-ivc) = (($\sigma$ ($\delta$18Oforam))2+($\sigma$ (Temp))2*(0.23)2)1/2

leading to calculated 2 $\sigma$ std of +/- 0.35 ‰ and +/- 0.68 ‰ for G. ruber w and G. truncatulinoides, respectively. Given the relative large error in temperature and $\delta$18Ow reconstructions, we only used 5-point average time series for interpretation to investigate longer-term trends in the datasets. The $\delta$18Ow records were corrected for ice volume, using the eustatic sea level curve of (Austermann et al., 2013) and are expressed as $\delta$18Ow-ivc values throughout the text. Changes in the AF front were reconstructed using the relative abundance of G. ruber w. published by (Weinelt et al., 2015). Abundance counts possess a 2 $\sigma$ std of +/- 2.5 %. As G. ruber w. is most abundant within the STG (Ottens, 1991; Schiebel et al., 2002; Storz et al., 2009), low/high abundances indicate a southward/northward movement of the AF relative to the coring site. Because the position of the AF is mainly related to changes in the westerly wind belt, the abundances of G. ruber w. indicate the relative contribution of subtropical water and can be used as tracer for the position of the northern STG rim and thus of the position of westerlies (see also argumentation in Repschläger et al., 2015) before. " Changes in

the AF front were reconstructed using the relative abundance of G. ruber w. published by (Weinelt et al., 2015) These abundance counts have an 2 $\sigma$ STD +/- 2.5 %. As G. ruber w. is most abundant within the STG (Ottens, 1991; Schiebel et al., 2002; Storz et al., 2009), low/high abundances indicate a southward/northward movement of the AF relative to the coring site. Because the position of the AF is related to changes in the westerly wind belt, the abundances of G. ruber w. indicate the relative contribution of subtropical water and can be used as tracer for the position of the northern STG rim and thus of the position of westerlies (see also argumentation in Repschläger et al., 2015). "

Additionally to these 2 major critisisms raised by Referee #2, we will perform all technical corrections and text passages suggested to further improve the manuscript. In a revised version of the manuscript, we will add the following references:

Austermann, J., Mitrovica, J. X., Latychev, K., and Milne, G. A.: Barbados-based estimate of ice volume at Last Glacial Maximum affected by subducted plate, Nature Geosci, 6, 553-557, 2013.

Bahr, A., Nürnberg, D., Karas, C., and Grützner, J.: Millennial-scale versus long-term dynamics in the surface and subsurface of the western North Atlantic Subtropical Gyre during Marine Isotope Stage 5, Global and Planetary Change, 111, 77-87, 2013.

Blaschek, M., Renssen, H., Kissel, C., and Thornalley, D.: Holocene North Atlantic Overturning in an atmosphere-ocean-sea ice model compared to proxy-based reconstructions, Paleoceanography, 30, 2015PA002828, 2015.

Cleroux, C., Cortijo, E., Anand, P., Labeyrie, L., Bassinot, F., Caillon, N., and Duplessy, J.-C.: Mg/Ca and Sr/Ca ratios in planktonic foraminifera: Proxies for upper water column temperature reconstruction, Paleoceanography, 23, PA3214, 2008.

Cleroux, C., Debret, M., Cortijo, E., Duplessy, J.-C., Dewilde, F., Reijmer, J., and Massei, N.: High-resolution sea surface reconstructions off Cape Hatteras over the last 10

ka, Paleoceanography, 27, PA1205, 2012.

Farmer, E. J., Chapman, M. R., and Andrews, J. E.: Holocene temperature evolution of the subpolar North Atlantic recorded in the Mg/Ca ratios of surface and thermocline dwelling planktonic foraminifers, Global and Planetary Change, 79, 234-243, 2011.

Martin, P. A. and Lea, D. W.: A simple evaluation of cleaning procedures on fossil benthic foraminiferal Mg/Ca, Geochemistry, Geophysics, Geosystems, 3, 8401, 2002.

Olsen, J., Anderson, N. J., and Knudsen, M. F.: Variability of the North Atlantic Oscillation over the past 5,200 years, Nature Geosci, 5, 808-812, 2012.

Ottens, J. J.: Planktic foraminifera as North Atlantic water mass indicators, Oceanologica Acta, 14, 123-140, 1991.

Repschläger, J., Weinelt, M., Kinkel, H., Andersen, N., Garbe-Schönberg, D., and Schwab, C.: Response of the subtropical North Atlantic surface hydrography on deglacial and Holocene AMOC changes, Paleoceanography, 30, 2015.

Schiebel, R., Schmuker, B., Alves, M., and Hemleben, C.: Tracking the Recent and late Pleistocene Azores front by the distribution of planktic foraminifers, Journal of Marine Systems, 37, 213-227, 2002.

Schwab, C., Kinkel, H., Weinelt, M., and Repschläger, J.: Coccolithophore paleoproductivity and ecology response to deglacial and Holocene changes in the Azores Current System, Paleoceanography, 27, PA3210, 2012.

Shackleton, N. J.: Attainment of isotopic equilibrium ocean water and the benthonic foraminifera genus Uvigerina: Isotopic changes in the ocean during the last glacial, Cent. Natl.Rech. Sci. Colloq. Int., 219, 203–209, 1974.

Solignac, S., de Vernal, A., and Hillaire-Marcel, C.: Holocene sea-surface conditions in the North Atlantic-contrasted trends and regimes in the western and eastern sectors (Labrador Sea vs. Iceland Basin), Quaternary Science Reviews, 23, 319-334, 2004.

Storz, D., Schulz, H., Waniek, J. J., Schulz-Bull, D. E., and Kuçera, M.: Seasonal and interannual variability of the planktic foraminiferal flux in the vicinity of the Azores Current, Deep Sea Research Part I: Oceanographic Research Papers, 56, 107-124, 2009.

Thornalley, D. J. R., Elderfield, H., and McCave, I. N.: Holocene oscillations in temperature and salinity of the surface subpolar North Atlantic, Nature, 457, 711-714, 2009.

Weinelt, M., Schwab, C., Kneisel, J., Hinz, M., and Climate and societal change in the western Mediterranean area around 4.2 ka BP. In: 2200 BC – Ein Klimasturz als Ursache für den Zerfall der Alten Welt? 2200 BC – A climatic breakdown as a cause for the collapse of the old world? 7. Mitteldeutscher Archäologentag vom 17. bis 19. Oktober 2013 in Halle (Saale). 7th Archaeological Conference of Central Germany October 23–26, 2014 in Halle (Saale). Meller, H., Arz, H. W., Jung, R., and Risch, R. (Eds.), Tagungen des Landesmuseums für Vorgeschichte Halle 12 Halle [Saale] 2015.

---

## Author Response (AR1)

**Response to the Reviews on the manuscript**
**"Holocene evolution of the North Atlantic subsurface transport"**
**by Janne Repschläger et al.**

Response to Referee #1

We greatly acknowledge the thoughtful scientific comments and very detailed technical hints for corrections in the text and figures of Referee #1. They improved the manuscript considerably.

Although in general positive towards our study, Referee #1 asked for a critical reconsideration of three major issues that are addressed in the revised version of the manuscript

1) Referee #1 proposed to include a brief discussion about the role of the strength of Mediterranean outflow water (MOW) on the position of the Azores Front / Azores Current (AF/AC), as MOW may exert a pivotal influence on the existence of the AC (Volkov et Fu 2010). We included a paragraph discussing the potential role of MOW for the AF/AC position.

2) Referee #1 proposed to include the Labrador Sea temperature and salinity records from P-012 and P-094 (Solignac et al., 2004) in one of our figures. We agree that the reasoning in the manuscript will strongly benefit from including theses records that are included in the revised version.

3) Referee #1 asked to specify size fractions of *G. ruber* w. and *G. truncatulinoides* as well as the coiling direction of *G. truncatulinoides* of the specimens used for analyses in order to avoid species specific and size related offsets in temperature reconstructions. Furthermore, under specific comments referee #1 asked to improve the methods section by providing more specific details. In order to address all methods related questions, we rewrote the methods chapter.

Additional specific comments by Referee #1 have been addressed as indicated in the following point-to point answer:

ABSTRACT Line 2, ": : : interactions with the subtropical gyre: : :" is quite unspecific.
Maybe replace by ": : : advection of saline water from the subtropical gyre". l.
    → *The sentence has been changed according to your suggestion*

11, "published data": specify which type of data (salinity and temperature).
    → *The data type has been added .*

l. 13/14, "Subsurfacewarm water transport started at about 8 ka with subtropical heat storage:" thissentence is again quite unspecific. Specify: What is the direction of the heat transport and what is the exact relation to heat storage?
    → *Sentence has been changed accordingly*

INTRODUCTION p. 1, l. 21: remove comma after "due to"
p. 1, l. 30: replace "ocean" with "oceanic"
p.2., l. 29: "entertained" is an odd phrasing here. Better replace with e.g. "discussed".
    → *the text has been changed according to your suggestion*

REGIONAL SETTING p. 3, l. 10 (and elsewhere): "G. ruber w.": "w." should not be in italics.
    → *the correct writing of G. ruber* w. *has been checked and corrected throughout the text.*

METHODS This section is quite superficial regarding the analytical details:

- pleasestate the number of individual forams picked for stable isotope and trace element analyses.
- were Mg/Ca measurements monitored for contamination by checking e.g. Al, Fe, Mn vs. Ca ratios?
- How were the samples cleaned? With or without reductive cleaning? These are quite elementary analytical information!
- Which machine was used for _18O and Mg/Ca analysis at which laboratory?
- It would be good to have a brief reference to the age model.
p. 3, l. 24 - : please state the equations used for Mg/Ca – temperature conversion explicitly.
p. 3, l. 26: "(Repschläger et al., 2015)" please check for the format of the citation (also in other parts of the text).
p. 4, l. 7: there are two ".." after "comparison".
p. 4, l. 9: instead of using the low-resolution sea level correction after Waelbroeck et al. (2002), a more recent sea level curve should be used such as Austermann et al. (2013)
p. 4, l. 9: check format of "Cleroux 2011"
p. 4., l. 10: explicitly state the equation (S8) from Cleroux et al. (2011) here.
p. 4, l. 11:"5- point": remove blank
p. 4, l. 13: Weinelt et al. (2015) is missing in the reference list. general: please introduce "w-ivc" as abbreviation. Note that the respective x-axis in Fig. 2C is labeled with "SW-IV".

> → *All missing information has been added to the methods chapter and the chapter has been corrected under consideration of you comments.*

RESULTS p. 4, l. 21: add ",respectively" after "8 ka BP".

> → *The text has been changed according to your suggestion*

p. 4, l. 22-23: the positive excursion in _18O: might there be a relation to the discontinuity mentioned later in the text?

> → *Yes, it is, also mentioned in the text, now.*

p. 4, l. 24 (and elsewhere): check for consistency that there is no blank before "_C"

> → *The spelling has been corrected throughout the text.*

p. 4, l. 29: "single points cannot be interpreted": this sentence is somewhat nonsensical as single points should be not interpreted in general.

> → *The part of the sentence was deleted.*

p. 4, l. 29: "indicate a evolution": insert a reference to Fig. 3 here.
p. 4, l. 30: there seems to be a number missing after "10.5 and"
p. 5, l. 1: there seems to be an "and" missing in "1.5 ‰ 1.7‰´'
p. 5, l. 10: insert an "in" after "any changes"

> → *The text has been corrected according to your suggestions.*

DISCUSSION p. 5, l. 12-13: as mentioned for the introduction there is no need to introduce the INTIMATE stratigraphy here.

> → *The sentence has been excluded from the discussion.*

p. 5, l. 25 (and Fig. 3): site ODP 1058 and core MD99-2203 are from subtropical waters, not tropical sites. Both are also not from the Caribbean as stated in the Fig. 3 caption. Better refer to western subtropical Atlantic.

> → *The text has been corrected according to your suggestions.*

p. 6, l. 3: delete comma after "above" p. 6, l. 5: insert "." after "trends"

> → *The text has been corrected according to your suggestion.*

p. 6, l. 7: "very similar" is an overstatement in my opinion
→ *The phasing has been changed to "similar"*
p. 6, l. 8: not sure what is meant by "12/10_C". Does this refer to the temperature range of Tsub?
→ *Yes, it does, the text has been changed to become more comprehensive.*

p. 7, l. 4:insert a blank before "(Repschläger et al., 2015)"
p. 7, l. 29-31: this paragraph is too speculative and should be omitted, unless the authors prove the existence of the 1500 yrs-cycle by spectral analysis.
→ *The paragraph has been deleted.*

p. 8, l. 10: the reference to "AMOC evolution" is somewhat misleading in this context as the authors reconstruct in the first place the dynamics of the STG and SPG.
→ *The latter is assumed to be linked to AMOC strength, however, the sentence has been changed.*

FIGURES Fig. 1: Indicate the position of core MD99-2203. Please place the circle indicating the position of ODP 1058 above the red arrow. The blue arrow is at places hard to see, a darker tone would help here.
→ *The Figure has been changed*

Fig. 2: In the caption a reference is missing to the source of the G. ruber abundance data (Weinelt et al., 2015). The readability of the caption would benefit if the listing of items is separated by comma (e.g. ": : :, c): : :" and a full stop at the end. In general, the error bars at individual points suggest that they represent individual replicates which is not the case. It would be better to state the 2sigma-error next to the respective y-axis.

Fig. 3: Please put "18"in _18O into superscript.
Fig. 4: The listing of items in the caption should be separated by comma
(e.g. ": : :, c): : :", also insert a full stop at the end.
→ *The figure captions have been changed according to your suggestions*

→ *The part has been replaced into the beginning of the discussion .*

Methods, page 4, lines 1-5: Explain in more detail
→ *The methods section has been rewriten under consideration of all your comments.*

Results, page 5, lines 3-8: Interpretations that belong to discussion
→ *The section has been added to the discussion*

Discussion, page 5, line 26: Consider to add core ID in order to facilitate reading of figure.
→*The core IDs have been added*

Discussion, page 6, line 6: The reference is to Figure 3b, but it is in fact Figure 3c?
→ *thanks, the number has been changed*

Discussion, page 6, lines 20-6: Unclear; explain in more detail.
→ *The discussion has been partially rewritten.*

Discussion, page 7, lines 6-31: Include additional studies and discuss all drivers in more detail
→ *The discussion has been rewritten under consideration of all comments*

Discussion, page 7, line 15: The acronym ITCZ needs to be defined.
→ *The definition has been added.*

References, page 8, line 27: Make sure that "K0.., N" reads "Koç, N" or "Koc, N" in final version.
→ *The reference has been controlled.*

Figure 1: Add positions and core ID for the studies from Labrador Sea.
→ *Missing core IDs and positions have been added.*

Figure 3: In the figure caption data are described and interpreted; this should be removed.
→ *All interpretation have been deleted from figure captions.*

Figure 4: The acronyms on the figure should be defined in the figure text in order to facilitate reading.
→ *Definitions of the acronyms were added to the figure caption*

**References**

[revised manuscript text omitted]